# `EtiCor`: Corpus for Analyzing LLMs for Etiquettes

**Ashutosh Dwivedi**[*]     **Pradhyumna Lavania**[*]     **Ashutosh Modi**
Indian Institute of Technology Kanpur (IIT Kanpur)
{ashutoshd20,pradhyumna20}@iitk.ac.in
ashutoshm@cse.iitk.ac.in

## Abstract

Etiquettes are an essential ingredient of day-to-day interactions among people. Moreover, etiquettes are region-specific, and etiquettes in one region might contradict those in other regions. In this paper, we propose **EtiCor**, an Etiquettes Corpus, having texts about social norms from five different regions across the globe. The corpus provides a test bed for evaluating LLMs for knowledge and understanding of region-specific etiquettes. Additionally, we propose the task of Etiquette Sensitivity. We experiment with state-of-the-art LLMs (Delphi, Falcon40B, and GPT-3.5). Initial results indicate that LLMs, mostly fail to understand etiquettes from regions from non-Western world.

## 1 Introduction

Etiquettes define a system of rules and conventions that regulate social and professional behavior. Etiquettes are an integral part of every culture and society. Consequently, etiquettes carry significant cultural and regional implications in every region of the world. While some of the social norms[1] are common across many cultures, every region has some society-specific norms that may be contradictory to norms in other societies. When a person is visiting another region/culture, they need to be aware of social norms in that region to avoid undesirable situations that may hamper the business and sometimes lead to conflicts.

In recent times, generative models and Large Language Models (LLMs) have become part of many digital technologies (such as web search and personal digital assistants (PDA)). In the past, people have referred to various books written to teach and explain etiquettes for a region (Hitchings, 2013; Martin, 2011; Vanderbilt, 1958; Post et al., 2011; Foster, 2002a,b). However, given the advances in

---

[*]Equal Contributions
[1]In this paper, we use the term etiquette and social norm inter-changeably

| Wikipedia Data | Percentage |
|---|---|
| English Articles among all articles | 76 |
| Active Editors (for English Articles) having Western Origin | 93 |

Table 1: Wikipedia statistics for English articles

technology, nowadays people usually refer to web-search and PDAs to inquire about social norms of the region they plan to visit. However, are LLMs — the underlying foundation of the technological mediums (e.g., search engines) used to glean social norms-specific information — sensitive to culture-specific etiquettes? Many LLM models have been developed using data (scrapped from the internet) heavily skewed toward Western cultures; for example, language models like BART (Lewis et al., 2019) are trained using Wikipedia text as the primary data source. However, an investigation of statistics of Wikipedia pages (Table 1) authored by various writers reveals that the English content is mainly created by people belonging to western (North America and Europe) societies (Yasseri et al., 2012). When authoring an article, people tend to be implicitly influenced by the culture they live in, and hence, this tends to introduce some culture/society-specific biases in the content. To promote research towards understanding how LLMs generalize concerning the understanding of knowledge about etiquettes of different cultures and regions and with the aim to analyze if the responses generated by the generative language models are skewed towards or against certain cultural norms and values, we introduce a new corpus **EtiCor** (Etiquettes Corpus). The corpus consists of texts on social norms followed in different world regions. The corpus is in English, but we plan to make it multi-lingual in the future. One of the focuses of **EtiCor** is to investigate potential biases LLMs exhibit when addressing regional etiquettes. The long-term goal of our research is to pave the

way for the development of AI systems that are more inclusive and sensitive to cultural differences. This paper introduces a corpus to enable the research community to advance toward the goal. We also propose the task of *Etiquette Sensitivity* to evaluate LLM's sensitivity towards different etiquettes. In a nutshell, we make the following contributions:

- We introduce a new corpus `EtiCor`. It is a corpus of texts about etiquettes of five major regions of the world. The corpus consists of 36k social norms in the form of labeled English sentences.

- As a use case for the corpus, we propose the task of *Etiquette Sensitivity*. Given a statement about a social norm about a region, the task of Etiquette Sensitivity is to classify if the social norm is appropriate concerning that region. We experiment with some of the existing state-of-the-art LLMs in zero-shot settings. Results indicate gaps in the knowledge of LLMs. Further, we experiment with a supervised setting, train a smaller transformer-based model (like BERT) on the corpus, and test its performance on the task. Fine-tuned BERT model performs well on the task of Etiquette Sensitivity. We release the dataset and code via the GitHub repository: https://github.com/Exploration-Lab/EtiCor.

## 2 Related Work

In recent years, there has been works in the NLP and ML communities to develop algorithms to understand human ethics and social commonsense (Jiang et al., 2021; Lourie et al., 2021; Forbes et al., 2020; Tolmeijer et al., 2020). We describe some of the major works in here. To develop an ethically sound ML model Jiang et al. (2021) have developed the Delphi system. The model is trained to perform three different tasks: a) classify (binary: yes/no) if a given social situation is as per social norms, b) open text judgment: whether the model agrees with the action taken in the particular situation or not c) relative selection: given a context and two actions, the task is to select the action that is more appropriate. The authors create a new dataset, Commonsense Norm Bank, that incorporates existing datasets such as Social Chemistry (Forbes et al., 2020), Scruples (Lourie et al., 2020), Ethics (Hendrycks et al., 2020), Commonsense Morality (Frederick, 2009), Moral Stories (Emelin et al., 2021), and Social Bias Inference Corpus (Sap et al., 2020). Delphi uses a pre-trained language model

| Region | #Social Norms |
|---|---|
| East Asia (EA) | 7432 |
| India (IN) | 4556 |
| Middle East and Africa (MEA) | 10031 |
| North America-Europe (NE) | 6156 |
| Latin America (LA) | 8172 |
| Total | 36347 |

Table 2: `EtiCor` distribution.

called Unicorn (based on the T5 model) (Lourie et al., 2021) to develop a system to judge whether a situation and a corresponding action are ethically correct and acceptable. An interesting study in grasping the moral context of a text is done via the Moral Integrity Corpus (MIC) (Ziems et al., 2022). Authors compiled prompt-reply pairs (prompts are questions about social norms) crowdsourced via websites like Ask-Reddit (https://www.reddit.com/r/AskReddit/). These prompt-reply pairs were further used to create a distinct 99k Rule of Thumb (RoT) having hard-coded rules based on social norms. LLM models (GPT-Neo and Dialo-GPT) are trained using these annotated RoTs to judge the sentiment and morality of the text. In another work, Abrams and Scheutz (2022) examines how pragmatic cues (in the form of social norms) affect the resolution of mentions in the task of conference resolution. In the study, different daily life situations are textually described, and the machine response is compared to the human response, giving an idea about how much social context is utilized by the model. A human-annotated database of social norms that apply to everyday tasks under various constraints is presented in NORMBANK (Ziems et al., 2023). The authors propose a SCENE framework that links a particular norm with surrounding attributes and constraints like (who/what/where) to include contextual information.

## 3 `EtiCor`: Corpus for Etiquettes

We create `EtiCor` (**Etiquettes Corpus**), a corpus of regions-specific etiquettes. In this work, we use a standard dictionary definition of etiquette and define it as a set of social norms/conventions or rules that tell how to behave in a particular social situation. Etiquettes are region-specific, dictate social and professional behavior, and are subjective. In the following text, we sometimes refer to each etiquette as a social norm or norm for short. The corpus contains specific etiquettes (as English sentences) relevant to a particular region, chosen

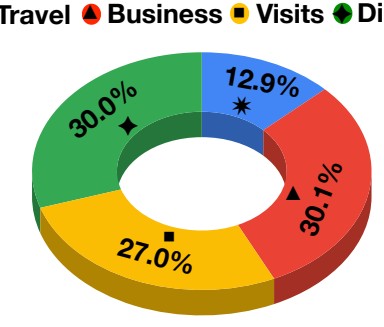

**Travel** ● **Business** ■ **Visits** ◆ **Dining**

12.9% ✳
30.1% ▲
27.0% ■
30.0% ◆

Figure 1: Etiquette Type Distribution

| Sentences | Labels | Regions |
|---|---|---|
| DON'T pour soy sauce directly over sushi or sashimi | 1 | EA |
| Pointing a chopstick towards someone while having dinner | -1 | EA |
| Women should not appear in front of non-family men without covering themselves | 1 | MEA |
| Eating with bare hands is common practice here | 1 | IN |
| Be sure to put any advanced educational degree and your full title or position on both sides of your business card. | 1 | LA |

Table 3: Few examples from `EtiCor`

from relevant sources to ensure authenticity. In `EtiCor`, we cover five regions across the globe: East Asia, India, the Middle East and Africa, North America and Europe, and Latin America. Table 2 shows the number of social norms for each region. The scope of norms was expansive as we took care that culture-specific practices relevant to a particular region were aptly included in the corpus, for example, festival-specific norms for India and Latin America, procedures related to religion in the Middle East and Africa. We also tried to collect data for generic day-to-day practices for each region to cover any cultural pattern engraved in daily routines like eating food, kitchen manners, etc. `EtiCor` covers four major types of etiquettes: 1. Dining and Festivals, 2. Visits and Social Interactions, 3. Travel, and 4. Business. Fig. 1 shows the distribution of etiquettes across types. App. Table 6 shows region-wise distribution of norms. A norm's average length (in terms of word count) is 18; App. Fig. 3 shows the word length histogram. Table 3 shows a few examples from `EtiCor`.

**`EtiCor` Creation:** To avoid conflicting information, we created the Etiquettes Corpus by collecting texts from authentic and publicly available sources (e.g., government-aided websites). The sources of information included websites providing information about regional etiquettes, tour guide points and pamphlets, etiquette information channels, and tweets and magazines on etiquettes (details about sources on GitHub repo). The collected (and scrapped) data was noisy and sometimes repetitive, so it was pre-processed. We manually removed texts that described the same norm; sentences were filtered if the length was less than 4 tokens, and large paragraphs were summarized. The pre-processed data were manually labeled.

**Labeling:** A norm in the corpus is labeled as acceptable (with label +1, positive class) if it is a general etiquette of the region, and it is labeled non-acceptable (with label -1, negative class) if it

is an act that is frowned upon by people of that culture. About 20% of the norms were self-labeled as these started with words like "Do" and "Do not." The remaining norms were manually labeled after careful examination. App. Table 7 provides the distribution of labels across different regions.

**Choice of Regions:** We selected regions based on diversity and data availability to cover as much of the world as possible. We could not include Russia, China, and other Southeast Asian countries due to the unavailability of a sizable number of social norms in the English language. We did not aim for automatic translations (to English) of social norms as these could not be verified for correctness.

**1) East Asia (EA):** We included the following countries: Japan, Korea, Taiwan, Vietnam, Malaysia, and the Philippines. These countries have several customs in common, especially dining and traveling; however, country-specific etiquettes are labeled with the country tag. This helped to prevent any conflict among the samples in the corpus. In this case, the corpus focuses mainly on dining, greeting, public transportation, and taboos.

**2) Middle East and Africa (MEA):** Since this is a large area with a diverse culture, we studied it carefully to capture various variations. We excluded some countries, such as Israel, Congo, and South Africa, where culture differed significantly from other regions. We also focused on common aspects of etiquette, such as dining, greetings, and shared religious norms. In cases of minor conflict, we included the country name or the cultural group with which it was associated.

**3) India (IN):** India is a diverse and culturally rich country where different societies co-exist. Moreover, India is the most populated country in the

| Region | Delphi | | | Falcon-40B | | | GPT-3.5 | | |
|---|---|---|---|---|---|---|---|---|---|
| | Accuracy | F1 score | # of Abstentions | Accuracy | F1 score | # of Abstentions | Accuracy | F1 score | # of Abstentions |
| EA | 0.63 | 0.69 | 809 | 0.67 | 0.70 | 480 | 0.55 | 0.67 | 32 |
| IN | 0.71 | 0.77 | 154 | 0.71 | 0.73 | 806 | 0.52 | 0.64 | 14 |
| MEA | 0.65 | 0.71 | 1149 | 0.69 | 0.73 | 123 | 0.55 | 0.68 | 46 |
| NE | 0.78 | 0.85 | 220 | 0.74 | 0.78 | 453 | 0.62 | 0.75 | 36 |
| LA | 0.69 | 0.71 | 1654 | 0.59 | 0.61 | 731 | 0.54 | 0.67 | 88 |
| Overall | 0.68 | 0.74 | 3986 | 0.67 | 0.71 | 2593 | 0.55 | 0.68 | 216 |

Table 4: Results of different models on `EtiCor`. The models are evaluated after removing the abstentions. Overall scores are calculated by a weighted sum of region-wise scores. Please refer to the text for more details.

world.[2] It was essential to prepare a common culture-oriented corpus. In India, several religious and social norms are practiced by Indians, as well as gift-giving and home-visiting-related practices. We also included some etiquette related to popular festivals.

**5) North America and Europe (NE):** We added cultural customs and practices from North America and Europe to the corpus. We grouped the two regions due to their strong correlation. Also, most of the etiquettes have their country tags attached to them to avoid any conflict.

**4) Latin America (LA):** With extensive area coverage in this region, we faced similar problems as in the case of the MEA region, and it was tackled similarly. Nevertheless, we covered several countries, and the only exceptions were Cuba and Colombia, which were removed because of the availability of a very small amount of data.

## 4 Etiquette Sensitivity

**Task Definition:** Existing LLMs are trained on huge corpora of texts, which has been shown to result in an improved overall general understanding of language. However, we are also interested in testing if the LLMs understand region-specific societal etiquettes. `EtiCor` serves this purpose. We propose the task of **Etiquette Sensitivity** to test our hypothesis. Given a statement about etiquette, the task is to predict whether the statement is appropriate for a region.

**Experiments:** We evaluated various LLMs in a zero-shot setting for the task. We experimented with Delphi (a pre-trained language model with 11B parameters) (Jiang et al., 2021), Falcon-40B (an open-source LLM with 40B parameters) (https://huggingface.co/blog/falcon) and GPT-3.5 Turbo (a closed LLM with 175B

parameters) (Brown et al., 2020). We analyzed the overall and etiquette type-wise performance of each of the models. We measured the model performance using standard accuracy and the F1-score metric. We also calculated overall scores by taking a weighted sum of the scores of each of the regions, where the weight for each region is the normalized count of number of instances for that region (more details in App. B). Each of the models was queried for etiquette appropriateness; a majority of times, a model replied in binary yes or no answer; in some instances, it provided an explanation that was manually mapped to labels +1 or -1. However, in some cases, the model did not provide an answer or gave an unclear response that could not be mapped to either of the labels. Such samples were not included in the evaluation and are referred to as *abstentions*. For checking Etiquette Sensitivityof Falcon-40B and GPT-3.5 models, we prompted each of the models to provide a single word (yes or no) answer. The prompt used was: "In a single word, say whether this is a good etiquette or socially acceptable norm in <region>:<etiquette-statement>."

**Results and Analysis:** Table 4 shows results for the three models. We also provide some example outputs in the App. Table 11. The models are evaluated after removing abstentions. We also calculated accuracy by including the abstentions; the results are shown in the App. Table 8. As can be observed from Table 4, scores for the North America-Europe (NE) region are much higher than for other regions. Also, the number of abstentions in the case of NE regions is much lower than in other regions. This is possibly due to the fact that these LLMs are primarily trained on data from the NE regions and, hence, are oblivious to etiquettes from other regions, pointing towards model bias for NE region etiquettes. Another observation from

---
[2]https://tinyurl.com/7ayaxdj2

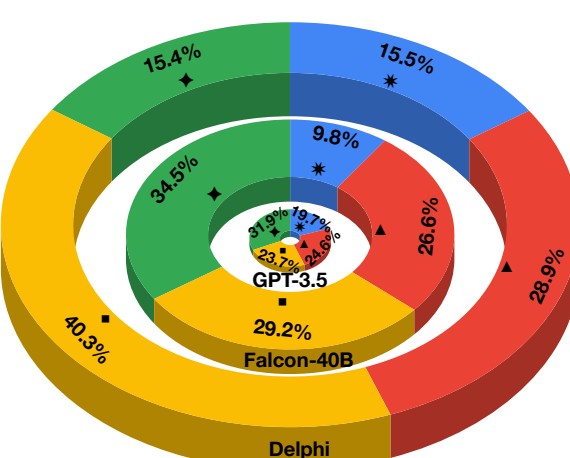

**Travel** ● **Business** ■ **Visits** ● **Dining**

Figure 2: **Percentage of Wrong Predictions** for each etiquette type by a model. The innermost circle results are for GPT-3.5, the middle circle is for Falcon-40B, and the outermost circle is for Delphi.

the results is that GPT-3.5 (despite being the largest model among the three models) has the worst performance for all regions. However, the number of abstentions for each region in the case of GPT-3.5 is the least among the three models, possibly because of its larger size and wider coverage. The results of the Falcon model are slightly worse than the Delphi model, possibly because the Delphi model is specifically trained with social norms as the end goal, and Falcon is a general-purpose LLM. Falcon also had a high abstention rate for the India (IN) region, where it stated a lack of regional knowledge as the primary reason for not answering. Further examination of the prediction outputs revealed that women-related etiquettes in the MEA region were misclassified the most. Overall, we saw similar trends across all three models.

We also analyzed the percentage of wrong predictions for each etiquette type for each model. Fig. 2 shows the percentage of wrong predictions for Delphi, Falcon-40B, and GPT-3.5. App. Table 9 provides detailed results. More details about the percentage of wrong prediction calculations are provided in the App. C. Region-wise percentage wrong predictions for each model are provided in the App. Table 10. As can be observed in Fig. 2, Travel and Business etiquette are predicted well by all the models due to the global nature of these etiquette types. Etiquettes related to Dining and Visits are usually region-specific; hence, the performance of models on these is poor. In the case of GPT-3.5, we also observed an increase in the contribution of wrong predictions in Business etiquette, which

| Region | Test Accuracy | Test F1 |
|---|---|---|
| EA | 0.872 (±0.034) | 0.913 (±0.038) |
| IN | 0.878 (±0.028) | 0.882 (±0.028) |
| MEA | 0.938 (±0.027) | 0.946 (±0.023) |
| NE | 0.919 (±0.032) | 0.934 (±0.032) |
| LA | 0.869 (±0.015) | 0.913 (±0.023) |
| Overall (Weighted) | 0.899 | 0.922 |
| Regions Combined | 0.871 (±0.015) | 0.902 (±0.021) |

Table 5: 5-fold Cross validation performance of fine-tuned BERT model for various regions and over the entire dataset (all regions combined).

may be related to changes in greeting styles and business format.

**Fine-tuned Models:** We also experimented with fine-tuned models for the task of Etiquette Sensitivity. In particular, we tried the BERT (Devlin et al., 2018) model. We fine-tuned BERT model on `EtiCor` (Train-Val-Test split=70:10:20 for each fold, ADAM learning_rate=0.001, batch=8 and 5 epochs). A separate model was trained for each region. The model was tuned to predict whether a given input is considered an appropriate etiquette in a particular region. The class imbalance was overcome by generating negative values of positive labels (Lee et al., 2021) and elimination of excess positive class by merging smaller sentences together. Furthermore, we did 5-fold cross-validation for each region as initial experiments showed variance in BERT's performance across different runs. We also ensured the proper distribution of each type of etiquette in all the folds so that any form of bias could be avoided easily. Table 5 shows the results averaged across 5-folds (more details in App. E). Overall results are calculated by a weighted sum of the mean scores for each region. We also combined the data from all the regions and performed 5-fold cross validation using BERT. The BERT model has fairly high performance for all regions, pointing out that region-specific training should be done when dealing with subjective and culture-specific norms.

## 5 Conclusion

In this paper, we presented `EtiCor`, a corpus of etiquettes covering major world regions. We further evaluated the performance of LLMs on the task of Etiquette Sensitivity and the results indicate a significant gap in knowledge and understanding of LLMs. In the future, we plan to develop region-specific Adapters and integrate them into an LLM via a mixture of experts (Wang et al., 2022).

## Limitations

In this paper, we proposed a new corpus and experimented on the task of Etiquette Sensitivity in a limited set of few LLMs. We do not develop any new model and leave it for future work. This resource paper aims to introduce the corpus and the task and show the limitations of LLMs when it comes to region-specific etiquettes. The work is a first step towards making more sophisticated etiquette-sensitive models.

## Ethical Considerations

Corpus was created from publicly available information, and no copyright was violated. During corpus creation, we ensured that any personal information was removed to prevent a model from developing any biases. To the best of our knowledge, we took all steps to keep the corpus as bias-free as possible and are unaware of any direct ethical consequences.

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

# Appendix

## A `EtiCor` Details

Table 6, Figure 3, and Table 7 provide more details about the corpus.

| Region | # Travel | # Dining | # Visits | # Buisness |
|--------|----------|----------|----------|------------|
| EA     | 1198     | 2605     | 1439     | 2190       |
| MEA    | 1041     | 2907     | 2861     | 3222       |
| IN     | 690      | 1305     | 1504     | 1057       |
| LA     | 1107     | 2184     | 2494     | 2387       |
| NE     | 642      | 1921     | 1508     | 2085       |
| Total  | 4678     | 10922    | 9806     | 10941      |

Table 6: Distribution of different etiquette types

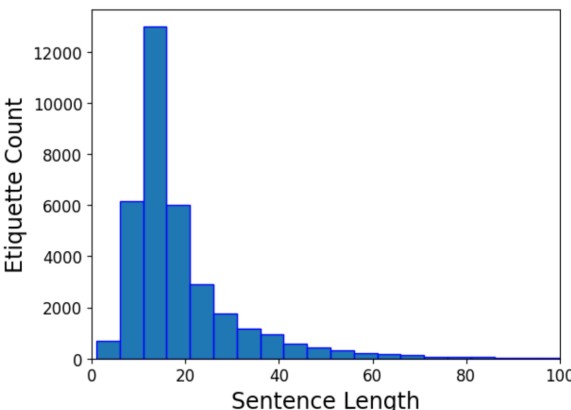

Figure 3: Word Length Distribution for `EtiCor`

| Region | # Positive Labels (+1) | # Negative Labels (-1) | Total |
|--------|------------------------|------------------------|-------|
| EA     | 4218                   | 3214                   | 7432  |
| MEA    | 5733                   | 4298                   | 10031 |
| IN     | 2356                   | 2200                   | 4556  |
| LA     | 4544                   | 3628                   | 8172  |
| NE     | 4230                   | 1926                   | 6156  |
| Total  | 21081                  | 15266                  | 36347 |

Table 7: `EtiCor`'s label distribution

## B Evaluation Metrics

$$\text{Precision} = \frac{TP}{TP + FP}$$

$$\text{Recall} = \frac{TP}{TP + FN}$$

$$F1 = \frac{2 \cdot \text{Precision} \cdot \text{Recall}}{\text{Precision} + \text{Recall}}$$

$$\text{Accuracy}_{\text{without Abstention}} = \frac{TP + TN}{TP + TN + FP + FN}$$

$$\text{Accuracy}_{\text{with Abstention}} = \frac{TP + TN}{\text{Total \# Data Points}}$$

where:

TP is the number of true positives (correct predictions of positive cases)

TN is the number of true negatives (correct predictions of negative cases)

FP is the number of false positives (incorrect predictions of positive cases)

FN is the number of false negatives (incorrect predictions of negative cases)

$$\text{Overall Score} = \sum_{r=1}^{N} w_r * Score_r,$$

where,

$w_r = \frac{count_r}{count_{total}}$, where, $count_r$ is the number of samples in region $r$ and $count_{total}$ is total number of samples in the dataset.

$Score_r$ is the accuracy/F1 score for region $r$.

## C Results Details

Table 8 provides results when abstained outputs are considered negative labels. In this case, we could not calculate the F1 score, as it is difficult to automatically classify an abstained output as a false positive or false negative. Table 9 shows the percentage of wrong predictions for each etiquette type for each model. These are calculated by taking all the incorrect predictions a model makes and then manually classifying them into various etiquette types. Subsequently, the ratio of wrong predictions for each type out of all the incorrect predictions gives the percentage of wrong predictions (for a model) for that type. Table 10 shows the percentage of wrong predictions for each model for each region.

## D Model Output Examples

Table 11 provides some example outputs of the models.

## E BERT Results

Table 12, 13, 14, 15, and Table 16 show 5-fold cross validation results for each of the region. We also combined the data from each of the region and performed 5-fold cross validation using the BERT model. The results are given in the last row of Table 5. Detailed results are shown in Table 17.

| Region | Delphi | | Falcon | | GPT3.5 | |
|---|---|---|---|---|---|---|
| | **Accuracy with Abstentions** | **# of Abstentions** | **Accuracy with Abstentions** | **# of Abstentions** | **Accuracy with Abstentions** | **# of Abstentions** |
| **EA** | 0.56 | 809 | 0.621 | 480 | 0.54 | 32 |
| **IN** | 0.69 | 154 | 0.603 | 806 | 0.52 | 14 |
| **MEA** | 0.57 | 1149 | 0.681 | 123 | 0.55 | 46 |
| **NE** | 0.75 | 220 | 0.697 | 453 | 0.62 | 36 |
| **LA** | 0.55 | 1654 | 0.541 | 731 | 0.53 | 88 |
| **Overall** | 0.61 | 3986 | 0.63 | 2593 | 0.55 | 216 |

Table 8: Results of models after including abstained predictions as negatives

| Etiquette Type | Delphi | Falcon-40B | GPT-3.5 |
|---|---|---|---|
| **Travel** | 15.46 | 9.82 | 19.70 |
| **Business** | 28.87 | 26.56 | 24.65 |
| **Visits** | 40.27 | 29.17 | 23.74 |
| **Dining** | 15.40 | 34.45 | 31.91 |
| **Overall** | 100.00 | 100.00 | 100.00 |

Table 9: **Percentage of Wrong Predictions** for each model for each etiquette type. Note that the numbers in the table are percentages.

| Etiquette Type | Delphi | | | | | Falcon-40B | | | | | GPT-3.5 | | | | |
|---|---|---|---|---|---|---|---|---|---|---|---|---|---|---|---|
| | EA | IN | MEA | NA | LA | EA | IN | MEA | NA | LA | EA | IN | MEA | NA | LA |
| **Travel** | 65.4 | 37.2 | 21.2 | 20.7 | 16.6 | 36.5 | 47.8 | 57.6 | 30.5 | 65.0 | 28.8 | 22.3 | 11.8 | 22.0 | 36.7 |
| **Business** | 12.4 | 35.3 | 27.7 | 42.3 | 11.0 | 23.3 | 13.8 | 19.5 | 19.3 | 11.3 | 18.9 | 46.3 | 12.6 | 14.8 | 40.5 |
| **Visits** | 12.2 | 12.2 | 24.1 | 25.6 | 56.2 | 26.8 | 11.3 | 10.8 | 30.3 | 10.5 | 36.7 | 21.0 | 26.7 | 21.1 | 10.1 |
| **Dining** | 10.0 | 15.3 | 27.0 | 11.4 | 16.2 | 13.4 | 27.1 | 12.1 | 19.9 | 13.2 | 15.6 | 10.4 | 48.9 | 42.1 | 12.7 |

Table 10: **Percentage of Wrong Predictions** for each etiquette type for each region by various models. Note that the numbers in the table are percentages.

| Region | Sentence | Gold Label | Delphi | Falcon-40B | GPT-3.5 |
|--------|----------|------------|--------|------------|---------|
| EA | English is widely spoken in business and government contexts though there are some exceptions such as trading. | -1 | 1 | 1 | 1 |
| EA | Pointing a chopstick towards someone while having dinner | -1 | 1 | 1 | -1 |
| EA | Public display of affection of opposite genders is fine | 1 | 1 | 1 | 1 |
| EA | Extended office hours are widely accepted throughout Japan. | 1 | -1 | -1 | 1 |
| IN | In rural India, it is a common practice to greet guests and invite them to sit on the floor | 1 | 1 | abstain | -1 |
| IN | Yawning in public is considered impolite you must cover your mouth when you yawn | -1 | 1 | 1 | 1 |
| IN | Eating with bare hands is common practice here | 1 | -1 | -1 | 1 |
| IN | Its okay to drink alcohol in parks and public gardens | -1 | -1 | -1 | -1 |
| IN | Indians often ask intrusive questions out of curiosity about foreign people. | 1 | -1 | 1 | 1 |
| IN | India is still a very conservative nation and hugging and kissing are not common practices especially with a newly made acquaintance | 1 | 1 | 1 | 1 |
| IN | Its unethical to eat beef in India with religious context added to it. | 1 | 1 | -1 | 1 |
| MEA | Women should be sensitive to the fact that they may be seated only with other women | 1 | -1 | -1 | -1 |
| MEA | Typically, conversation while eating is extensive, and most Nigerians avoid eye contact when dining | -1 | 1 | 1 | abstain |
| MEA | Follow the guidelines and procedures for performing Tawaf AlIfadah and the symbolic stoning of the devil Ramy al Jamarat during Hajj | 1 | 1 | 1 | 1 |
| MEA | Use of Perfumes Fragrances play a crucial part in Kuwaiti society It's not uncommon to be offered perfume at social gatherings | 1 | -1 | 1 | 1 |
| MEA | Mawtayn or marhabtayn alan is a less formal greeting | 1 | -1 | abstain | -1 |
| MEA | In some of the countries in the Arab world video taping is illegal and traditionally, in Islam images of people and the human form are considered sacrilegious | 1 | -1 | -1 | -1 |
| MEA | Ethiopians shake hands slowly when greeting each other | -1 | 1 | 1 | 1 |
| NE | In Hungary: Close friends kiss one another lightly on both cheeks, starting with the left cheek. | 1 | -1 | -1 | 1 |
| NE | Avoid excessive physical contact, such as hugging, unless it is culturally inappropriate | 1 | 1 | -1 | 1 |
| NE | Keep a fire blanket or baking soda quite far away to prevent small kitchen fires. | -1 | 1 | -1 | 1 |
| LA | Restaurants usually have the percent tip already included on the bill. | 1 | -1 | -1 | 1 |
| LA | Children in Colombia are expected to be respectful and not overly conversational when speaking with adults | -1 | 1 | 1 | 1 |
| LA | There is a rigid separation of the genders in Haiti, more so in the less urbanised areas. Women mainly sell the crops and wares in the market while men work the fields and at manufacturing jobs in small shops | 1 | -1 | -1 | 1 |

Table 11: Some Examples of Etiquette's and their corresponding zero shot results

| Fold | Validation Accuracy | Test Accuracy | Test F1 Score |
|---|---|---|---|
| Fold 1 | 0.869 | 0.836 | 0.873 |
| Fold 2 | 0.879 | 0.859 | 0.900 |
| Fold 3 | 0.880 | 0.869 | 0.908 |
| Fold 4 | 0.902 | 0.869 | 0.907 |
| Fold 5 | 0.950 | 0.928 | 0.978 |
| Overall | 0.896 (±0.032) | 0.872 (±0.034) | 0.913 (±0.038) |

Table 12: BERT Model: validation accuracy, test accuracy, and test F1 score for each fold for EA region.

| Fold | Validation Accuracy | Test Accuracy | Test F1 Score |
|---|---|---|---|
| Fold 1 | 0.918 | 0.892 | 0.880 |
| Fold 2 | 0.902 | 0.887 | 0.890 |
| Fold 3 | 0.863 | 0.842 | 0.854 |
| Fold 4 | 0.939 | 0.903 | 0.908 |
| Fold 5 | 0.881 | 0.864 | 0.876 |
| Overall | 0.901 (±0.028) | 0.878 (±0.028) | 0.882 (±0.028) |

Table 13: BERT Model: validation accuracy, test accuracy, and test F1 score for each fold for IN region.

| Fold | Validation Accuracy | Test Accuracy | Test F1 Score |
|---|---|---|---|
| Fold 1 | 0.939 | 0.918 | 0.928 |
| Fold 2 | 0.979 | 0.965 | 0.970 |
| Fold 3 | 0.980 | 0.969 | 0.973 |
| Fold 4 | 0.927 | 0.919 | 0.928 |
| Fold 5 | 0.939 | 0.916 | 0.931 |
| Overall | 0.953 (±0.025) | 0.938 (±0.027) | 0.946 (±0.023) |

Table 14: BERT Model: validation accuracy, test accuracy, and test F1 score for each fold for MEA region.

| Fold | Validation Accuracy | Test Accuracy | Test F1 Score |
|---|---|---|---|
| Fold 1 | 0.868 | 0.858 | 0.878 |
| Fold 2 | 0.958 | 0.930 | 0.950 |
| Fold 3 | 0.964 | 0.938 | 0.955 |
| Fold 4 | 0.927 | 0.918 | 0.945 |
| Fold 5 | 0.936 | 0.920 | 0.942 |
| Overall | 0.931 (±0.038) | 0.919 (±0.032) | 0.934 (±0.032) |

Table 15: BERT Model: validation accuracy, test accuracy, and test F1 score for each fold for NE region.

| Fold | Validation Accuracy | Test Accuracy | Test F1 Score |
|---|---|---|---|
| Fold 1 | 0.935 | 0.880 | 0.909 |
| Fold 2 | 0.941 | 0.858 | 0.878 |
| Fold 3 | 0.943 | 0.875 | 0.919 |
| Fold 4 | 0.917 | 0.848 | 0.918 |
| Fold 5 | 0.914 | 0.882 | 0.942 |
| Overall | 0.930 (±0.013) | 0.869 (±0.015) | 0.913 (±0.023) |

Table 16: BERT Model: validation accuracy, test accuracy, and test F1 score for each fold for LA region.

| Fold | Validation Accuracy | Test Accuracy | Test F1 Score |
|---|---|---|---|
| Fold 1 | 0.911 | 0.890 | 0.915 |
| Fold 2 | 0.904 | 0.883 | 0.921 |
| Fold 3 | 0.881 | 0.847 | 0.864 |
| Fold 4 | 0.897 | 0.873 | 0.914 |
| Fold 5 | 0.879 | 0.866 | 0.895 |
| Overall | 0.894 (±0.012) | 0.871 (±0.015) | 0.902 (±0.021) |

Table 17: BERT Model: validation accuracy, test accuracy, and test F1 score for each fold for data of all regions combined.