# OpenReview forum: "EtiCor: Corpus for Analyzing LLMs for Etiquettes"
_EMNLP/2023/Conference — EMNLP 2023 Main_

### Official Review · Reviewer_FAuQ · 2023-07-30

**Soundness:** 4

**Excitement:**

4: Strong: This paper deepens the understanding of some phenomenon or lowers the barriers to an existing research direction.

**Paper Topic And Main Contributions:**

The paper presents EtiCor, a novel English corpus of etiquettes covering five major world regions. More precisely, the corpus includes data for five major regions: India, Latin America, Japan-Korea-Taiwan, Middle-East-Africa, North America, and Europe. The corpus has been constructed by scraping websites providing information about regional etiquettes, tour guide points, pamphlets, etiquette information channels tweets and books on etiquettes. The scraped norms have been manually labeled as pertaining to a specific region (with label +1) or not belonging to the specific region (with label -1).
Finally, the authors introduced a novel Etiquette Sensitivity task: given a statement about etiquette, the task is to predict whether the statement is appropriate for a region or not. Three Large Language Models (LLMs) have been assessed on the novel task and the results show no particular difference among the assessed models (Delphi, Falcon and GPT3.5). Additionally, the authors fine-tuned four different BERT binary classifiers reporting small improvement over the LLMs assessed in a zero-shot setting.

**Questions For The Authors:**

1. The related work section focuses on ethically sound Machine Learning (ML) models, is there any work related to corpus on social norms?
2. In the discussion of the preprocessing step (line 168), two filters on sentence length have been mentioned but never clarified. What is the range of sentence lengths accepted?
3. In the discussion of results achieved by LLMs the authors mention the fact that "scores are low for each region". May the authors argue on this point? In my opinion the F1 scores are not that low, especially when compared with the fine-tuned BERT model on the dataset. Although the numbers are not comparable, BERT scores little better than the three LLMs.
	3.A. Did the authors consider framing the task as multi-label? Reducing the problem to a binary task is a strong simplification; it might be interesting to study the performance of the models in a case where they have to decide for which region a norm is acceptable.

**Reasons To Accept:**

The topic is interesting, the paper is well-written and easy to follow. The paper introduces a novel corpus with intense manual curation.

**Reasons To Reject:**

The paper is not always self-contained: for example no general definition of etiquette has been provided, although it can be guessed from the text, I suggest the authors provide a clear definition from the introduction. The experimentation on the Etiquette Sensitivity task should be reviewed: the results achieved by the BERT model are not comparable to those reported on Table 4 since they have been obtained on different partitions of the same set. In order to make these numbers comparable a ten-fold cross validation approach should be used for the fine-tuned system.

**Reproducibility:**

3: Could reproduce the results with some difficulty. The settings of parameters are underspecified or subjectively determined; the training/evaluation data are not widely available.

**Reviewer Confidence:**

3: Pretty sure, but there's a chance I missed something. Although I have a good feel for this area in general, I did not carefully check the paper's details, e.g., the math, experimental design, or novelty.

**Typos Grammar Style And Presentation Improvements:**

Please find some minor comments below:
- please consider providing a definition for personal digital assistants on line 032;
- please consider referencing Table 2 before Table 6 on rows 152-153, this may help the reader to understand the abbreviations in tables that have been introduced in Table 2;
- please consider removing the two "and" on line 163, rewriting the list as regional etiquettes, tour guide points, pamphlets, etiquette information channels tweets and books on etiquettes;
- please consider introducing vertical lines on Table 4 separating the three different LLMs and introducing the bold notation for highest model for each region;
- please consider rewriting Sensitivityby on line 279.

---

> ### Author Rebuttal · Authors · 2023-08-28
>
> Thanks for your insightful review and thoughtful suggestions.
>
> ---
>
> Thanks for pointing out the missing definition of etiquette. We use a standard dictionary definition of an etiquette and define it as a set of social norms/conventions or set of rules which tell about how to behave in a particular social situation. If accepted, we will update the paper to include the definition of etiquette.
>
> ---
>
> Thanks for pointing out that zero-shot results of LLMs and BERT are not comparable. Yes, we agree with this idea and we will perform BERT experiments using ten-fold cross-validation.
>
> ---
>
> **Questions**
> 1. Related Work: Yes, to the best of our knowledge, there are two corpora related to social norms (SocialChem and Moral Integrity Corpus). We have written about this in lines 118-130. Due to space limitations we could not include more details but if accepted we can include more details.
> 2. Range of sentence length: minimum length was 50 characters, any sentence having length below 50 characters was removed..
> 3. The accuracy and F1 score reported in Table-4 are for cases where models always reported a definite answer (i.e., without abstention cases). If we had considered abstention as incorrect prediction then the overall accuracy and F1 scores comes out much lower (by approx. 15-20%). If accepted, we include results with abstention in the paper as well.
> 4. Thanks for pointing out the idea of posing a problem as multi-label. However, there are some etiquette which are contradictory to each other in different regions ( lines 193-104 and lines 204-206), so this can create problems in the multi-label setting. Nevertheless, we could remove such conflicting information and then we can try out the multi-label setting and provide results in the updated version of the paper.
>
> ---
>
> Thanks for pointing out typos and minor mistakes, we will fix these in the updated version of the paper.

---

### Official Review · Reviewer_j5Th · 2023-07-31

**Soundness:** 4

**Excitement:**

4: Strong: This paper deepens the understanding of some phenomenon or lowers the barriers to an existing research direction.

**Paper Topic And Main Contributions:**

This paper introduces an interesting topic, by analyzing the (gap of) knowledge of LLM about localized etiquettes. It also reveals how LLM are biased towards Western societies, missing many aspects of other cultures, such as etiquette.

They propose EtiCOR, a corpus of 35k annotated social norms/ etiquettes followed in 5 big world regions, namely India, Latin America,
Japan-Korea-Taiwan, Middle-East-Africa, North America, and Europe. The corpus is in English, but the objective is to make it multilingual in the future.

The authors also propose the task of Etiquette Sensitivity to test several models. Given a statement about a social norm pertaining to a region, the objective is to classify if the social norm is appropriate or not relative to that region. They experiment with zero-shot learning with LLM (i.e. Delphi, Falcon-40B and GPT3.5, and then compared it with a finetuned BERT model.
The paper addresses the analysis of the potential biases regarding social etiquettes included in LLM, with the final goal of contributing towards more responsible and inclusive AI systems.



**Reasons To Accept:**

This is a well-written, easy-to-follow and straightforwardly explained paper, which I personally appreciate.

The authors use norms extracted from verified sources, such as guides, government web pages, etc. They only use information in English. I believe it's a good point to avoid automatic translation systems as these can have the same biases regarding social etiquette as LLM, so the translation could not be accurate. I don't know if the decision was taken considering that, but it's a strong point to back it.

The annotation is by region, but in specific cases, the label can point to a single country or even a specific cultural group. I consider this dataset very interesting and useful, not only for NLP purposes but also to better understand others and create a more equal, understanding and responsible society in general, I will be very interested in having a look at it once it is made available.

The GPT3.5 results about MEA social norms, which the model refuses to process and/or qualifies as gender-specific hate are very interesting and could motivate an extensive multidisciplinary qualitative analysis.

Table 9 examples of outputs are very interesting, one of the most interesting tables in the paper, but it's not referred to in the whole paper (it is in app.C, but many readers might not arrive there! Consider either including some examples in the main body of the paper, or at least letting the reader know that that table is in the Appendix because it's really interesting.)

**Reasons To Reject:**

Falcon40B results are confusing, consider re-writing.

Experiments with BERT, are the results coming from a single run? There is much difference between validation and test accuracy, which points to either a different distribution in both chunks or a high degree of randomness in the results. Running several runs and calculating the average would make the results much stronger.



**Reproducibility:**

3: Could reproduce the results with some difficulty. The settings of parameters are underspecified or subjectively determined; the training/evaluation data are not widely available.

**Reviewer Confidence:**

4: Quite sure. I tried to check the important points carefully. It's unlikely, though conceivable, that I missed something that should affect my ratings.

**Typos Grammar Style And Presentation Improvements:**

lines 150-151 - Dining and Festivals, Visits, and Social, Travel, and Business. --> The distribution of the commas makes it a bit confusing to understand the 4 groups. I assume it is 1) Dining and Festivals, 2) Visits and Social (?), 3) Travel and 4) Business, is that correct?

lines 152,155, 179, 256, 272 (at least) - You mention App. without saying in which appendix we can find the information, the appendix name / letter is missing.

lines 184 -187: Sentence "We could include China and other South-East Asian countries due unavailability of a large number of social norms in the English language" --> We could NOT include China [...] due TO unavailability [...]

It would be useful if you include the acronym for each region, so it's easier to read the tables.

line 261 --> You say scores are low for each region, but they don't seem that low, consider rephrasing.

lines 270-271 -->  "The results were quite close to Delphi but not good". --> consider rephrasing

In the results, you mention GPT3.5 first and ChatGPT later. Maybe it's better to unify the way of naming the model or explain that you use ChatGPT with GPT3.5.

---

> ### Author Rebuttal · Authors · 2023-08-28
>
> Thanks for the detailed and insightful review. We find your review very encouraging! Thanks! We are glad that you appreciate the simplicity of the paper and find experimental results interesting. Due to space constraint we had to move the examples table in Appendix, if accepted, we will move a part of it into the main paper.
>
> Thanks for pointing out confusion about Falcon-40B results, we will re-write and make it more clear.
>
> Regarding BERT, we ran it only once. Yes, we agree with you that we should report averaged out results over multiple runs. If accepted, we will include these results in the paper.
>
> Thanks for pointing out the typos, we will fix these.

---

### Official Review · Reviewer_vwjx · 2023-08-05

**Soundness:** 3

**Ethical Concerns:**

Yes

**Excitement:**

3: Ambivalent: It has merits (e.g., it reports state-of-the-art results, the idea is nice), but there are key weaknesses (e.g., it describes incremental work), and it can significantly benefit from another round of revision. However, I won't object to accepting it if my co-reviewers champion it.

**Missing References:**

- [NormBank: A Knowledge Bank of Situational Social Norms](https://aclanthology.org/2023.acl-long.429) (Ziems et al., ACL 2023)
- [Social Norms Guide Reference Resolution](https://aclanthology.org/2022.naacl-main.1) (Abrams & Scheutz, NAACL 2022)

**Paper Topic And Main Contributions:**

Authors present a dataset, EtiCor, containing text samples labelled for weather or not the text is an acceptable etiquette. Samples are sourced from diverse geographical locations. Author use the dataset as a test bed for evaluating LLMs for ability to be sensitive for to acceptable and unacceptable social norms (etiquettes).

**Questions For The Authors:**

- What is the theoretical underpinnings for definition of "Etiquette" and how do you ensure that dataset isn't subjective?
- What was the criteria for
     - annotating the data? or weak labelling?
     - selecting websites?
- What was the prompt used to query LLMs?

**Reasons To Accept:**

- Authors present a new corpus about etiquette of five regions of the world, contianing 35k labels.
- Based on the dataset, author propose a new task - Etiquette Sensitivity - to test if a social norm is appropriate or not given a geographical region.

**Reasons To Reject:**

- If there are existing resources to test model's response are ethical or stick to moral integrity - it is not clear the research gap a new etiquette corpus is filling. Including a discussion on the same would be immensely beneficial to the paper.
- Data creation methodology is presented without necessary details -
    - what is criteria that makes a sentence acceptable social norm or not?
    - what is the criteria for selecting those specific websites listed in appendix?
    - verification/validation of the scrapped data and if the labels were infact correct? How many annotators, guidelines for annotation?
    - Social norm can be very subjective - within a sub-geographic region as well - what is the rationale for combining continents/countries.
- Manner of querying models for their knowledge of social etiquette is not presented in the paper.

**Reproducibility:**

3: Could reproduce the results with some difficulty. The settings of parameters are underspecified or subjectively determined; the training/evaluation data are not widely available.

**Reviewer Confidence:**

4: Quite sure. I tried to check the important points carefully. It's unlikely, though conceivable, that I missed something that should affect my ratings.

**Typos Grammar Style And Presentation Improvements:**

- Appendix D can be much more readable if it can be condensed into a table.
- Appendix figures, tables and subsections can be referenced better. In its current form they are hard to follow.
- Some of the assumptions made in the paper - either should be supported with citations or shown emperically - for instance the claim that the model is trained on "data from North American and European region" (L259).
- Maybe short paper is not suitable for this work - as paper lacks details in most of the sections of the paper.

---

> ### Author Rebuttal · Authors · 2023-08-28
>
> We would like to thank you for your insightful comments and we are glad you appreciate the corpus. We provide detailed answers to your comments below:
>
> ---
> **Comparison to Existing Resources:**
>
> To the best of our knowledge there is no existing dataset that focuses on Etiquettes. We take a general dictionary definition as a set of social norms/conventions or set of rules which tell about how to behave in a particular social situation.
> The motivation behind EtiCor is to promote development of systems (e.g., LLMs) that are sensitive to etiquettes (social norms) of a region (as discussed in lines 70-74). Corpora created with regard to Ethics are usually based on ethical theories and do not necessarily address etiquettes and social conventions.  In fact, we tried an existing model (Delphi) trained on a ethics corpus (Rainbow corpus) and we found that its performance on EtiCor (Etiquette corpus) is not good (Table 4), pointing towards the gap in Delphi’s knowledge and hence there is a need for a corpus focusing on etiquettes.
> We discuss and compare with existing corpora in Section 2 (Related Work). Due to the space constraint of 4 pages we could go into more details but in the final version we can provide more details about differences between our corpus and corpora based on Ethics.
>
>  **Missing References:**
> Thanks for pointing missing references
> 1. We were not aware of NormBank work as it was released in July 2023 and we submitted the paper in June 2023. NormsBank corpus is about situational commonsense and norms and it comes close to our corpus. We will include details about this in our updated version of the paper.
> 2. Paper by Abrams & Scheutz, 2022 focuses on general etiquettes and not region specific etiquettes. Our corpus focuses on region specific etiquettes, as social norms vary across regions. Moreover, their study is mostly about the context of a sentence, and they have tried out different models to see how well they understood the context of a sentence, which helped them understand social norms. Our study on the other hand does not dwell much into the underlying linguistic context of the sentence and aims to include regional context by sensitizing about regional etiquettes.
>
> ---
> **Data Creation Methodology**
>
> 1. As described in the paper (lines 157-164), we scraped data from authentic, trusted and publicly available official travel and government websites which are trusted (e.g., by travelers), so each etiquette (sentence) is an acceptable social norm in that region.
> 2. Criteria for selection of website is already mentioned in lines 160-165.
> We didn’t involve annotators but ourselves validated the data. We also manually filtered out noisy sentences (line 167-170). We took each of the scraped sentences as it is and verified that it made sense.
> 3. Criteria for combining sub-geographic regions was based on geographical proximity of regions and variance in the etiquette of each of the sub-region. We also provide details in the paper (lines 182-225). We manually examined etiquette for each sub-region and then merged the regions that had similar etiquettes into a bigger region. We can provide more details about this in the updated version of the paper.
> ---
>
> We provided details about the prompt used for LLMs in lines 280-285. In the updated version of the paper, we can provide more details.
>
> ---
>
> **Questions**
> 1. We use the following definition of etiquette: set of social norms/conventions or set of rules which tell about how to behave in a particular social situation. By definition, since etiquette is region specific, it will be subjective.
> 2. We provide details about the criteria, labeling and website selection above.
> 3. We provide details about the prompt above.
>
> ---
>
> Thanks for your suggestions about typos, grammar and presentation style, we will fix these in the final version of the paper.

---

### Meta-Review · Area_Chair_BZMk · 2023-09-17

**Recommendation:** 5

**Metareview:**

**Summary:**
The authors introduce the EtiCor dataset, comprising 35,000 manually annotated text samples labeled for determining whether the text adheres to acceptable localized etiquette or not. Specifically, this corpus encompasses etiquettes from five major world regions plus Europe: India, Latin America, Japan-Korea-Taiwan, Middle-East-Africa, North America, and Europe. The dataset was constructed by scraping websites that provide information about regional etiquettes, tourist guide points, pamphlets, etiquette information channels, tweets, and books on etiquettes. While the corpus is currently in English, the long-term goal is to make it multilingual. The authors employ this dataset as a testbed for evaluating Large Language Models (LLMs) such as Delphi, Falcon-40B, and GPT3.5, introducing a novel Etiquette Sensitivity task. In this task, given a statement about etiquette, the goal is to predict whether the statement is appropriate for a specific region. This research reveals how LLMs exhibit biases toward Western societies while overlooking many aspects of other cultures. Additionally, the authors fine-tune four distinct BERT binary classifiers, reporting modest improvements over the LLMs evaluated in a zero-shot setting.

**Strengths:**
The reviewers are in agreement regarding the strengths of the paper:
1. The authors introduce a novel corpus focusing on etiquettes from five distinct global regions, featuring 35,000 meticulously curated labels.
2. Building upon this dataset, the authors propose a new task, 'Etiquette Sensitivity,' aimed at assessing the appropriateness of a social norm within a given geographical context.
3. The paper delves into an analysis of potential biases within Large Language Models (LLMs) concerning social etiquettes, with the overarching aim of promoting responsible and inclusive AI systems.
4. The dataset is not only valuable for NLP applications but also for fostering cross-cultural understanding and promoting a more equitable, empathetic, and responsible society.
5. Additionally, one reviewer highlights the interesting findings related to GPT3.5's responses to Middle East and Africa (MEA) social norms, where the model either refuses to process or categorizes them as gender-specific hate. This outcome could motivate an extensive multidisciplinary qualitative analysis.

**Weaknesses:**
Reviewers have identified the following weaknesses in the paper:
1. The paper lacks clarity in articulating the specific research gap addressed by the new etiquette corpus, as it may not be evident considering the existence of other resources for evaluating model responses in terms of ethical and moral integrity.
2. The methodology used to create the data is presented without the necessary level of detail.
3. The paper does not adequately describe the approach taken to query models for their knowledge of social etiquette.
4. Some of the results reported by the authors are not well-described, and the paper is not consistently self-contained. For instance, the results related to Falcon40B lack sufficient clarity, and there are discrepancies in the reported accuracy of BERT experiments between the validation and test sets. Furthermore, the BERT experiments do not employ a ten-fold cross-validation approach for the fine-tuned system.

**Author-Reviewer discussion and acknowledgment:**
The authors gave clarifications in response to the concerns raised by the reviewers and have outlined the planned improvements to be made during the rebuttal response and discussion phase. All reviewers have responded and acknowledged the authors' arguments

**Conclusion:**
The topic presented in this work is interesting, and the paper is well-written and easy to follow. However, reviewers suggest that the authors address the identified typos. Furthermore, reviewers recommend that the authors incorporate additional references and enhance the paper based on the points raised during the discussion phase.

---

### Decision · Program_Chairs · 2023-10-07

**Decision:**

Accept-Main

**Comment:**

**Summary:**
The authors introduce the EtiCor dataset, comprising 35,000 manually annotated text samples labeled for determining whether the text adheres to acceptable localized etiquette or not. Specifically, this corpus encompasses etiquettes from five major world regions plus Europe: India, Latin America, Japan-Korea-Taiwan, Middle-East-Africa, North America, and Europe. The dataset was constructed by scraping websites that provide information about regional etiquettes, tourist guide points, pamphlets, etiquette information channels, tweets, and books on etiquettes. While the corpus is currently in English, the long-term goal is to make it multilingual. The authors employ this dataset as a testbed for evaluating Large Language Models (LLMs) such as Delphi, Falcon-40B, and GPT3.5, introducing a novel Etiquette Sensitivity task. In this task, given a statement about etiquette, the goal is to predict whether the statement is appropriate for a specific region. This research reveals how LLMs exhibit biases toward Western societies while overlooking many aspects of other cultures. Additionally, the authors fine-tune four distinct BERT binary classifiers, reporting modest improvements over the LLMs evaluated in a zero-shot setting.

**Strengths:**
The reviewers are in agreement regarding the strengths of the paper:
1. The authors introduce a novel corpus focusing on etiquettes from five distinct global regions, featuring 35,000 meticulously curated labels.
2. Building upon this dataset, the authors propose a new task, 'Etiquette Sensitivity,' aimed at assessing the appropriateness of a social norm within a given geographical context.
3. The paper delves into an analysis of potential biases within Large Language Models (LLMs) concerning social etiquettes, with the overarching aim of promoting responsible and inclusive AI systems.
4. The dataset is not only valuable for NLP applications but also for fostering cross-cultural understanding and promoting a more equitable, empathetic, and responsible society.
5. Additionally, one reviewer highlights the interesting findings related to GPT3.5's responses to Middle East and Africa (MEA) social norms, where the model either refuses to process or categorizes them as gender-specific hate. This outcome could motivate an extensive multidisciplinary qualitative analysis.

**Weaknesses:**
Reviewers have identified the following weaknesses in the paper:
1. The paper lacks clarity in articulating the specific research gap addressed by the new etiquette corpus, as it may not be evident considering the existence of other resources for evaluating model responses in terms of ethical and moral integrity.
2. The methodology used to create the data is presented without the necessary level of detail.
3. The paper does not adequately describe the approach taken to query models for their knowledge of social etiquette.
4. Some of the results reported by the authors are not well-described, and the paper is not consistently self-contained. For instance, the results related to Falcon40B lack sufficient clarity, and there are discrepancies in the reported accuracy of BERT experiments between the validation and test sets. Furthermore, the BERT experiments do not employ a ten-fold cross-validation approach for the fine-tuned system.

**Author-Reviewer discussion and acknowledgment:**
The authors gave clarifications in response to the concerns raised by the reviewers and have outlined the planned improvements to be made during the rebuttal response and discussion phase. All reviewers have responded and acknowledged the authors' arguments

**Conclusion:**
The topic presented in this work is interesting, and the paper is well-written and easy to follow. However, reviewers suggest that the authors address the identified typos. Furthermore, reviewers recommend that the authors incorporate additional references and enhance the paper based on the points raised during the discussion phase.